# Proteomic and Metabolomic Profiles of T Cell-Derived Exosomes Isolated from Human Plasma

**DOI:** 10.3390/cells11121965

**Published:** 2022-06-18

**Authors:** Aneta Zebrowska, Karol Jelonek, Sujan Mondal, Marta Gawin, Katarzyna Mrowiec, Piotr Widłak, Theresa Whiteside, Monika Pietrowska

**Affiliations:** 1Maria Sklodowska-Curie National Research Institute of Oncology, 44-102 Gliwice, Poland; aneta7zebrowska@gmail.com (A.Z.); karol.jelonek@io.gliwice.pl (K.J.); marta.gawin@io.gliwice.pl (M.G.); katarzyna.mrowiec@io.gliwice.pl (K.M.); 2UPMC Hillman Cancer Center, University of Pittsburgh Cancer Institute, Pittsburgh, PA 15232, USA; mondals@upmc.edu; 3Clinical Research Support Centre, Medical University of Gdańsk, 80-210 Gdańsk, Poland; 4Department of Pathology, University of Pittsburgh School of Medicine, Pittsburgh, PA 15261, USA; 5Department of Medicine, University of Pittsburgh School of Medicine, Pittsburgh, PA 15261, USA

**Keywords:** CD3 antigen, exosomes, immune capture, T lymphocytes, metabolomics, proteomics, small extracellular vesicles

## Abstract

Exosomes that are released by T cells are key messengers involved in immune regulation. However, the molecular profiling of these vesicles, which is necessary for understanding their functions, requires their isolation from a very heterogeneous mixture of extracellular vesicles that are present in the human plasma. It has been shown that exosomes that are produced by T cells could be isolated from plasma by immune capture using antibodies that target the CD3 antigen, which is a key component of the TCR complex that is present in all T lymphocytes. Here, we demonstrate that CD3(+) exosomes that are isolated from plasma can be used for high-throughput molecular profiling using proteomics and metabolomics tools. This profiling allowed for the identification of proteins and metabolites that differentiated the CD3(+) from the CD3(−) exosome fractions that were present in the plasma of healthy donors. Importantly, the proteins and metabolites that accumulated in the CD3(+) vesicles reflected the known molecular features of T lymphocytes. Hence, CD3(+) exosomes that are isolated from human plasma by immune capture could serve as a “T cell biopsy”.

## 1. Introduction

Exosomes are small extracellular vesicles (sEVs) that are sized between 30–150 nm. They are produced by all types of cells via the endosome pathway and are present in all body fluids, including plasma, urine, cerebrospinal fluid, synovial fluid and breast milk [1,2,3]. The molecular and genetic cargo of sEVs reflects the content of their parent cells and thus, exosomes are considered to be promising components of “liquid biopsy”. Exosomes are key mediators in different aspects of cell-to-cell communication, including those involved in disease-related mechanisms. Tumor cells produce and release large numbers of sEVs, which are also referred to as tumor-derived exosomes or TEXs [4,5,6,7]. However, vesicles in human plasma are a heterogeneous mix of circulating sEVs that originated from multiple tissues, including immune cells [6,7,8]. Consequently, the EV component of plasma consists of many individual subsets of exosomes that share a common biogenesis but have unique phenotypic/functional characteristics. This heterogeneity causes difficulties in understanding exosome-mediated intercellular crosstalk in vivo without attributing their molecular/functional features to specific subsets of sEVs that are produced by various tissues or circulating parental cells. To be able to determine the molecular signatures of the different subsets of exosomes, novel strategies are required for their isolation and separation from human plasma for downstream molecular/genetic profiling. An emerging approach is the separation of particular tissue-derived sEVs based on their specific antigens using an immune capture strategy, as we recently reported for the case of sEVs in the plasma of melanoma patients [9,10,11].

Exosomes that are released by T cells, which comprise a large fraction of the sEVs in human plasma [12], are key messengers between tissue cells, malignant tumors and the immune system [13,14,15,16]. Therefore, the isolation of T cell-derived exosomes may result in substantial knowledge being gained about the crosstalk between immune cells and tissue-resident normal or pathologically altered cells. We previously reported on the success of an immune capture strategy that relies on the use of a specific mAb for the CD3 antigen (a component of the TCR signaling complex), which separates CD3(+) T cell-derived exosomes from CD3(−) exosomes that are released by other immune or non-immune cells [12,15,17]. These exosome fractions have been characterized functionally, which has revealed their important immunomodulatory role in patients with head and neck cancers [12]. Here, the same immune capture strategy for CD3(+) exosomes was utilized for the in-depth characterization of the differences between the proteome and metabolome compositions of exosomes that are released by T cells versus other types of CD3(−) exosomes that are present in the plasma of healthy individuals. We found that under physiologically normal conditions, the immune-captured CD3(+) exosomes reflected the proteomic and metabolomic features of their parental T cells and thus, could serve as a “liquid T cell biopsy”.

## 2. Materials and Methods

### 2.1. Isolation of Total sEVs from Human Plasma

Blood samples were obtained from 10 consenting healthy donors (HDs) (IRB approval #04-001). The blood samples were processed to separate the plasma, which was divided into aliquots and stored at −80 °C until thawed and was then used for the exosome isolation. The thawed and pre-cleared plasma was processed by ultrafiltration, followed by size exclusion chromatography (SEC) as previously described in [9]. Briefly, the thawed plasma samples were centrifuged at 2000 × *g* for 10 min, followed by centrifugation at 10,000 × *g* for 30 min at 4 °C and they were then ultrafiltered through 0.22-μm filters (EMD Millipore, Billerica, MA, USA). An aliquot (1 mL) of plasma was loaded onto a 10-cm SEC column and 1 mL fractions were eluted with PBS. The void volume fraction #4, which contained the majority of the non-aggregated and morphologically intact sEVs, was collected and used for further analyses. The transmission electron microscopy (TEM), vesicle size range, particle numbers and protein content of fraction #4 were determined as previously described in [9,18,19]. The sEV protein concentration was determined using the BCA method (Pierce Biotechnology, Rockford, CA, USA), as per the manufacturer’s instructions. The sEVs were concentrated using Vivaspin 500 (100,000 MWCO, Sartorius, Göttingen, Germany).

### 2.2. Isolation of CD3(+) Exosomes Using Immune Capture

The T cell-derived exosomes (CD3(+) exo) were separated from the non-T cell-derived exosomes (CD3(−) exo) using immune capture with anti-CD3 mAbs, which recognize an epitope that is selectively expressed on T cell receptor-positive (TCR+) T cells [12,15]. An aliquot of the sEVs that were present in fraction #4 (10 μg of protein) was used for the immune capture by biotin-labeled anti-CD3 mAbs (Biolegend #300304, San Diego, CA, USA) and streptavidin-labeled magnetic beads (ExoCap™, MBL International, Woburn, MA, USA). The vesicles were incubated with the biotin-labeled anti-CD3 mAbs overnight and then 100 μL of streptavidin-coated magnetic beads (washed twice with PBS) were added, which was followed by overnight incubation. The recovered CD3(+) vesicles that were captured by the anti-CD3 mAbs on the beads were washed twice with PBS and re-suspended in 100 μL of PBS as the CD3(+)exo fraction. Exosomes that were not captured on the beads, i.e., the soluble CD3(−)exo fraction, were also harvested. The detection of proteins that were present on the surface of the CD3(+)exo and CD3(−)exo fractions was performed using on-bead flow cytometry, as previously described in [12]. The separated exosome fractions were used for the downstream analyses.

### 2.3. Sample Preparation for Metabolomics and Proteomics Analyses

Sterile PBS (350 µL) was added to the thawed samples of CD3(+)exo on the beads, vortexed for 30 sec and then mixed (50 rpm) using a HulaMixer (HulaMixer™ Sample Mixer, Thermo Fisher Scientific, Waltham, MA, USA) for 15 min at 4 °C. The samples of the non-captured CD3(−)exo fraction were vortexed and centrifuged; then, each sample was adjusted to the final volume of 350 µL using sterile PBS. All samples containing 350 µL of suspension were transferred to new 2-mL Eppendorf tubes (in the samples containing beads with CD3(+)exo, a brown precipitate was visible). Extraction with ice-cold 100% MeOH was performed using vigorous vortexing for 1 min (the final MeOH concentration was 80%); then, the samples were mixed using a HulaMixer (50 rpm) for 10 min at 4 °C and centrifuged for 10 min at 14,000× *g* and 4 °C. The supernatants were collected into new tubes for metabolomics analysis, while all pellets were frozen at −20 °C for proteomics analysis. The supernatants were vacuum-concentrated using a SpeedVac concentrator (SpeedVac DNA 120, SAVANT Instruments, Inc., Ramsey, MN, USA) in 500 µL aliquots to reach the final remaining sample volume of 50 to 70 µL and were then stored at −80 °C until further processing.

### 2.4. Targeted Metabolomics Analysis

The methanol-extracted samples (see paragraph above) were analyzed using a targeted quantitative approach with an Absolute IDQ p400 HR kit (test plates in the 96-well format; Biocrates Life Sciences AG, Innsbruck, Austria), according to the manufacturer’s protocol. The samples were applied to the wells in a few 10–20 µL aliquots (dried under nitrogen) and were then analyzed using combined direct flow injection (for lipids) and liquid chromatography (for small metabolites) high-resolution mass spectrometry (HR-MS). The method combined the derivatization and extraction of the analytes with selective mass-spectrometric detection using integrated isotope-labeled internal standards for absolute quantification. This approach hypothetically allowed for the simultaneous quantification of 407 metabolites (or their isomer groups) into 42 amino acids and biogenic amines, 55 acylcarnitines, 60 di- and triglycerides, 196 (lyso)phosphatidylcholines, 40 sphingolipids, 14 cholesteryl esters and hexose. The mass spectrometry analyses were carried out on an Orbitrap Q Exactive Plus (Thermo Fisher Scientific, Waltham, MA, USA), which was equipped with a 1290 Infinity UHPLC (Agilent, Santa Clara, CA, USA) system that was controlled by Xcalibur 4.1 software (Thermo Fisher Scientific, Waltham, MA, USA). The acquired data were processed using Xcalibur 4.1 and MetIDQ DB110-2976 (Biocrates Life Sciences AG, Innsbruck, Austria) software.

### 2.5. Peptide Preparation for Proteomics Analysis

The pellets that were collected during the sample preparation (see paragraph above) were dissolved in 100 µL of lysis buffer (0.1 M Tris–HCl pH 8.0, 0.1 M DTT, 4% SDS), heated for 1 h at 99 °C with shaking (800 rpm) and then cooled down. The samples were subsequently centrifuged at 20,000× *g* for 10 min at RT; then, the supernatants were collected and subjected to filter-aided sample preparation (FASP) [20] using a Microcon-30 kDa Centrifugal Filter Unit with an Ultracel-30 membrane (Millipore, Billerica, MA, USA). The proteins that were retained on the membrane were alkylated using 50 mM of iodoacetamide and digested with sequencing grade modified trypsin (Promega, Madison, WI, USA) at an enzyme to protein ratio of 1:50 (m/m). The digestion was performed in a humid chamber at 37 °C for 18 h. The obtained tryptic peptides were released from the filter membrane using 160 µL of water, acidified with trifluoroacetic acid (final concentration of TFA: 0.2% *v*/*v*) and desalted using StageTips [21], which contained an Empore C18 SPE extraction disk (Supelco, Bellefonte, PA, USA). The peptides that were retained on the sorbent were eluted with 60% ACN and 0.1% TFA, dried using a vacuum concentrator and resolved in 20 µL of water; then, the peptide concentration was assessed using the tryptophan fluorescence method [22]. Before the LC-MS/MS analysis, the purified peptide samples were acidified with TFA (final concentration: 0.1% *v*/*v*).

### 2.6. Protein Identification by LC-MS/MS Analysis

The LC-MS/MS analysis of the tryptic peptides (see paragraph above) was performed using the Dionex UltiMate 3000 RSLC nanoLC system coupled with a Q Exactive Plus Orbitrap mass spectrometer (Thermo Fisher Scientific, Waltham, MA, USA). The peptides were separated on a reverse-phase Acclaim PepMap RSLC nanoViper C18 column (75 μm × 25 cm, 2 μm granulation) using 0.1% FA in LC-MS grade water (as mobile phase A) and 80% acetonitrile with 0.1% FA in LC-MS grade water (as mobile phase B) at 30 °C and a flow rate of 300 nL/min (for 200 min). For additional desalting purposes, the samples were loaded onto a C18 trap column for 3 min using 0.1% FA in LC-MS grade water as a loading buffer. After desalting, the trap column was switched with the analytical column and the peptides were eluted using the binary gradients of 3–8% of mobile phase B for 7 min, 8–35% of mobile phase B for 130 min and 35–60% of mobile phase B for a further 20 min. Finally, the rinsing off of the column in 80% of mobile phase B for 20 min and equilibration in 3% of mobile phase B for another 20 min were performed. The spectrometer was operated in data-dependent MS/MS mode with survey scans that were acquired at a resolution of 70,000 at *m*/*z* 50 in MS mode and 17,500 at *m*/*z* 200 in MS2 mode. The spectra were recorded using the positive ion scanning mode in the range of 350–1500 *m*/*z* and higher energy collisional dissociation (HCD) was used to fragment the ions.

The protein identification was performed using a reviewed Swiss-Prot human database (release 2018_11_30, which contains 11,378,269 sequence entries) with a precision tolerance of 10 ppm for the peptide masses and 0.02 Da for the fragment ion masses. All raw data that were obtained for each dataset were imported into Proteome Discoverer v.1.4 (Thermo Fisher Scientific, Waltham, MA, USA) <Thermo raw files> for protein identification and quantification (Sequest engine was used for the database searches). Protein was considered as positively identified when at least two peptides per protein were found by the search engine and the peptide score reached the significance threshold of FDR = 0.01 (assessed by the Percolator algorithm). A protein was further considered as “present” when it was detected in at least one sample of a given type. The abundance of the identified proteins was estimated in Proteome Discoverer using the Precursor Ions Area detector node, which calculates the abundance of a given protein based on the average intensity of the three most intensively distinct peptides for that protein with further normalization to the total ion current (TIC).

### 2.7. Statistical Analyses

The significance of difference of the levels of proteins/metabolites that was used in the quantitative analyses (compounds with less than 50% of the initial “zero” values in each group were used in a given comparison) was measured using the Wilcoxon signed-rank test. Additionally, the chi-squared independence test was applied to test whether the absence/presence status of a given compound was a group-related feature. The FDR correction was applied using the Benjamini–Hochberg procedure, when necessary. All statistical hypotheses were tested at the 5% significance level. The STRINGdb database [23] was used to predict the relationships between the chosen proteins.

## 3. Results

### 3.1. Separation of CD3(+) and CD3(−) Vesicles

The total populations of sEVs were isolated from the plasma of healthy donors using size-exclusion chromatography (SEC) and then separated into T cell-derived sEVs (CD3(+)exo) and other cell-derived sEVs (CD3(−)exo) using the immune capture method with anti-CD3 monoclonal antibodies. The total sEVs that were isolated from plasma by SEC (fraction #4) were characterized according to the MISEV2018 guidelines [24]. Morphology, size and the presence of endocytic protein markers (as well as the absence of cytoplasmic proteins) indicated that the majority of isolated sEVs represented exosomes. Figure 1A documents typical characteristics of isolated sEVs. The separation of the exosomes into the CD3(+) and CD3(−) fractions was monitored by on-bead flow cytometry, which revealed the enrichment of CD3 antigens in the CD3(+)exo fraction and the lack of CD3 antigens in the CD3(−)exo fraction (Figure 1B). We concluded that the combination of SEC for the isolation of the total plasma sEVs with the morphological and molecular characteristics of exosomes followed by the immune capture method with anti-CD3 mAbs allowed for the isolation of exosomes that were released by T lymphocytes and their separation from exosomes that were produced by other cells. The protein and lipid profiles of the resulting exosome fractions were assessed by mass spectrometry for 10 donors and the abundance of each identified component was compared in paired CD3(+)exo and CD3(−)exo vesicles from the same donor.

### 3.2. Comparison of the Protein Contents of CD3(+) and CD3(−) Vesicles

Using a shotgun proteomics approach, 418 proteins were identified (listed in the Appendix A), including 99 high-abundance plasma proteins that usually co-purify with plasma sEVs [25]. These putative plasma “contaminants” were excluded from all further analyses of the sEV components and were addressed separately. The quantitative analysis of the sEVs revealed several proteins that had an abundance that was significantly different (FDR < 0.05) in the CD3(+)exo and CD3(−)exo fractions (Figure 2A, left). We found 36 sEV proteins that were upregulated in the CD3(+)exo fraction and 56 sEV proteins that were upregulated in the CD3(−)exo fraction. On the other hand, almost half of the putative plasma proteins were upregulated in the CD3(−)exo fraction. This observation suggested that some of the plasma proteins that were putatively co-purifying with total sEVs (fraction #4) were removed from the CD3(+)exo fraction during the washing of the bead-captured vesicles.

Moreover, we compared the set of proteins that were identified in the CD3(+) exosomes to a set of proteins that were detected in T lymphocytes. We used the proteomics dataset that was provided by Joshi et al. [26], who performed an in-depth analysis of CD3^+^/CD4^+^/CD8^−^ T cells and identified 6572 proteins. We found that the majority of the protein characteristics for the CD3(+)exo fraction, i.e., neither upregulated in the CD3(−)exo nor the putative plasma components, were also detected in the T lymphocytes (Appendix A, Figure 2A, right). On the other hand, the protein characteristics for the CD3(+)exo fraction that were not detected in the CD3^+^/CD4^+^/CD8^−^ T cells (92 proteins) were mostly associated with exosome-based transport and putatively represented components that are specific to extracellular vesicles (Appendix A). It is noteworthy that a few reports have addressed functions of sEVs that are produced by different classes of T lymphocytes, including CD4^+^ cells (sEVs mediate co-stimulatory functions), CD8^+^ cells (sEVs from activated cells mediate suppressive functions), Treg cells (sEVs are strongly immunosuppressive) [14,27,28]. There are no data available on sEVs that are produced by naïve or memory T cells. However, none of the abovementioned studies comprehensively addressed the proteome composition of sEVs that are released by T lymphocytes. Therefore, our proteomics data that were obtained with sEVs that were produced in vivo by the overall population of T lymphocytes could not be compared to other proteomics datasets in this study.

In the next step, we identified the biological functions/processes that were associated with the differentially expressed proteins (DEPs) that were upregulated in either the CD3(+)exo or CD3(−)exo fractions. The complete lists of the overrepresented functions and processes are provided in the Appendix A. The potential interactions among the DEPs that were specific to both fractions of vesicles are also illustrated in the Appendix A, Figure 2B,C. We found that among the most abundant subsets of proteins that were upregulated in either the CD3(+)exo or CD3(−)exo fractions, there were proteins that were associated with immune-related processes (GO:0002376; 20 and 26 DEPs were upregulated in the CD3(+) and CD3(−) fractions, respectively) and the stress response (GO:0006950; 20 and 28 DEPs in the CD3(+) and CD3(−) fractions, respectively). Moreover, among the significantly overrepresented processes that were associated with the proteins that were upregulated in the CD3(−)exo fraction, there was “signaling” (GO:0023052; 31 DEPs). It is noteworthy, however, that the immune-related proteins that were upregulated in the CD3(+)exo and CD3(−)exo fractions of the plasma sEVs were associated with different types of immune cells. The immune-related proteins that were upregulated in the CD3(+)exo vesicles were primarily associated with leukocytes (GO:0045321 and GO:0050900; 17 DEPs were associated with “leukocyte activation” or “leukocyte migration”). On the other hand, the immune-related proteins that were upregulated in the CD3(−)exo fraction were primarily associated with neutrophiles (GO:0043312; nine DEPs were associated with “neutrophil degranulation”). Furthermore, a large subset of the proteins that were upregulated in the CD3(−)exo fraction was associated with platelets (GO:0030168 and GO:0002576; 11 DEPs were associated with “platelet activation” and “platelet degranulation”). Hence, the functions that were associated with the proteins that were upregulated in the two analyzed fractions of plasma exosomes confirmed their origin from T lymphocytes, which carried CD3 antigens (CD3(+)exo fraction) and other types of cells (CD3(−)exo fraction), including platelets and neutrophils.

### 3.3. Comparison of the Lipid and Small Metabolites Content of CD3(+) and CD3(−) Vesicles

In the second type of analysis, 338 metabolites (lipids and small metabolites) were identified and quantified using high-resolution mass spectrometry, including 287 putative membrane components (di/triglycerides, phosphatidylcholines, sphingolipids and cholesteryl esters), as well as a few acylcarnitines, amino acids, biogenic amines and hexoses (all compounds are listed in the Appendix A). The analysis revealed that several metabolites had an abundance that was significantly different (FDR < 0.05) between the CD3(+)exo and CD3(−)exo fractions. There were 96 metabolites that were upregulated in the CD3(+)exo fraction and 74 metabolites that were upregulated in the CD3(−)exo fraction of the plasma sEVs (Figure 3A). The majority of metabolites that were detected in the analyzed sEVs were lipids and lipid-related compounds. When small metabolites were considered, only hexoses (including glucose and fructose), which were highly accumulated in the CD3(+)exo fraction, discriminated between the fractions of the plasma exosomes.

The lipidomic profile characteristics for the CD3(+) and CD3(−) fractions of the plasma exosomes were identified. When the putative components of the vesicle membranes were analyzed, higher total amounts of cholesterols and sphingomyelins (SM) were observed in the CD3(+)exo fraction, while higher total amounts of phosphatidylcholines (PC) were observed in the CD3(−)exo fraction. In the case of acylglycerols, we noted higher amounts of triglycerides (TG) in the CD3(+)exo fraction, while we noted higher amounts of diglycerides (DG) in the CD3(−)exo fraction. Moreover, higher total levels of ceramides and acylcarnitines were characteristic for the CD3(+)exo fraction (Figure 3B). It was shown that the plasma membranes of lymphocytes were relatively enriched with cholesterol and sphingomyelins but depleted in phosphatidylcholines [29] and that cholesterols and sphingolipids were essential components of the plasma membranes that were involved in the proper functioning of the T cells [30]. Hence, the lipid composition of the CD3(+) sEVs that was revealed in the present study reflected the features of the plasma membranes of the T cells. Furthermore, ceramides (another class of metabolites that accumulated in the CD3(+) sEVs) were critical mediators that were associated with different functions of the T cells [31]. Similarly, the high concentration of glucose in the CD3(+) sEVs seemed to reflect a very high demand for this compound in the activated T cells [32]. Therefore, it should be noted that the features of the metabolic profiles that discriminated the CD3(+)exo fraction from the CD3(−)exo fraction of the plasma vesicles reflected the composition of the plasma membrane and other metabolic features of the T cells.

## 4. Conclusions

The exosomes that were released by T cells could be effectively separated from other types of sEVs that were present in human plasma using the immune capture method with antibodies that were specific for the CD3 antigen, which is a key component of the TCR signaling complex and is exclusively present in all subpopulations of T lymphocytes. The isolated and immunoselected vesicles represented a feasible material for high-throughput molecular profiling using proteomics and metabolomics, which allowed for the identification of the proteins and metabolites that differentiated the CD3(+) and CD3(−) fractions of the exosomes in the plasma of healthy donors. Importantly, the proteins and metabolites that accumulated in the CD3(+) vesicles reflected the known molecular features of T cells. Moreover, the protein characteristics for the CD3(+) vesicles were detected in the CD3^+^ lymphocytes. Hence, the exosomes that were purified from human plasma using the immune capture method with anti-CD3 mAbs appeared to serve as a “T cell biopsy”. Importantly, the discrimination of the exosome subsets in the plasma of HDs could provide a basis for future investigations on the CD3(+) exosomes in the plasma of patients with pathological conditions, including autoimmune diseases or cancers. A “T cell biopsy” using the exosomes from pathological plasma could replace the currently used analyses of T lymphocytes from blood or other body fluids. In addition, the CD3(−)exo fraction, which putatively reflected the attributes of other circulating or tissue-infiltrating immune and non-immune cells, could inform us as to their general activation or functional status.

## Figures and Tables

**Figure 1 cells-11-01965-f001:**
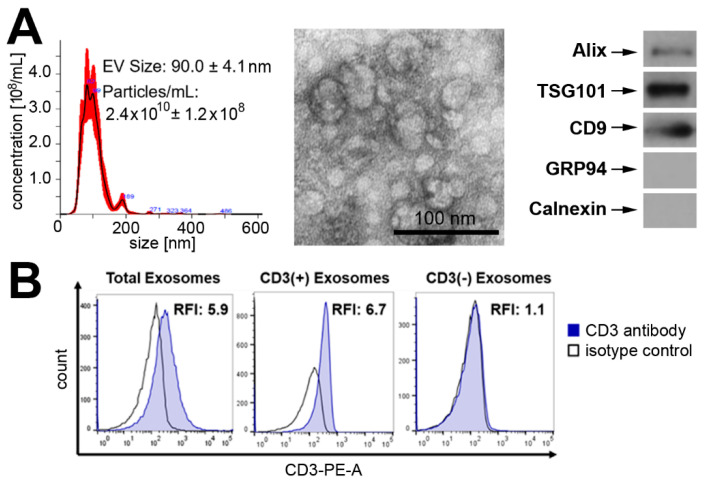
The characteristics of the analyzed vesicles: (**A**) the size, morphology and presence of exosome markers (from left to right), as analyzed by NanoSight, TEM and Western blots, respectively, in the total sEVs that were purified from the plasma; (**B**) the presence of CD3 in the CD3(+)exo and CD3(−)exo fractions, as analyzed by on-bead flow cytometry. RFI, MFI sample/MFI isotype control; TCR, T cell receptor CD3.

**Figure 2 cells-11-01965-f002:**
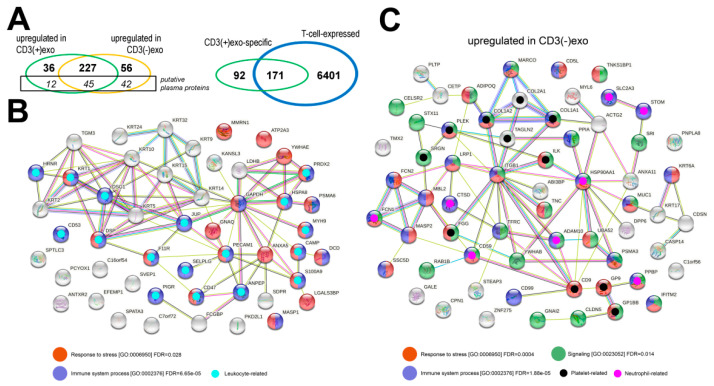
The proteins that were identified in the CD3(+) and CD3(−) fractions of the sEVs from human plasma: (**A**, **left**) a Venn diagram showing the numbers of proteins that were significantly upregulated in either fraction (FDR < 5%) with putative plasma proteins shown separately; (**A**, **right**) a Venn diagram showing the overlap between the CD(3+)exo-specific proteins and the T cell-expressed proteins (the latter dataset was taken from [26]); (**B**) the functional networks of proteins that were upregulated in the CD3(+)exo fraction; (**C**) the functional networks of proteins that were upregulated in the CD3(−)exo fraction. Proteins that were associated with the selected biological processes are color-coded, along with the significance of the process overrepresentation. The putative interactions between the proteins and associated processes were found using the STRINGdb database.

**Figure 3 cells-11-01965-f003:**
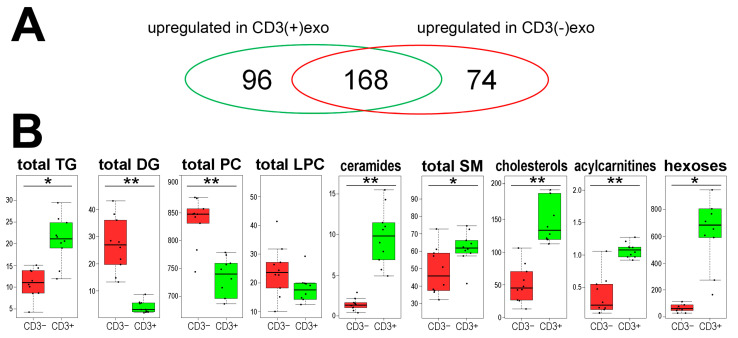
The metabolites that were identified in the CD3(+) and CD3(−) fractions of the sEVs from human plasma: (**A**) a Venn diagram showing the numbers of metabolites that were significantly upregulated in either fraction (FDR < 5%); (**B**) the abundance of the major classes of metabolites that were detected in the CD3(+) and CD3(−) fractions of the sEVs. The aggregated amounts of the major classes of lipids are also shown (TG, triglycerides; DG, diglycerides; PC, phosphatidylcholines; LPC, lysophosphatidylcholines; SM, sphingomyelins). The box plots represent the minimum, lower quartile, median, upper quartile and maximum. The dots represent the individual samples. The significance of difference between both fractions of sEVs is marked with asterisks (* FDR < 0.05; ** FDR < 0.001).

## Data Availability

Not applicable.

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
