# Peer review of "Proteomic and Metabolomic Profiles of T Cell-Derived Exosomes Isolated from Human Plasma"

_cells, 2022, doi:10.3390/cells11121965_

Round 1
Reviewer 1 Report
The paper titled “Proteomic and metabolomic profiles of T cell-derived exosomes 2 isolated from human plasma” by Zebrowska et al is aimed at studying T cell-derived, CD3+ sEV.
The exosomes were isolated by ultrafiltration followed by size exclusion chromatography, and then sorted based on CD3 expression on the surface. sEV were then characterized for metabolic and proteome analyzes.
The manuscript is well written and the data are well described. However, there are few minor issues that should be addressed.
It should be noted and discussed in the paper that the sampling of total T cell population in the circulation is expected to approach a mixture of populations, probably mostly naïve cells due to their abundance. Are there any published data on sEV production that compare naïve, memory and activated T cells?
The authors describe several T cell “function” that are differentially expressed between the two groups of sEV. It would be helpful to mention if any specific, typical proteins will be mentioned as well, for additional validation of the data.
Altogether this manuscript is a proof of concept for differential analysis of Sorted, T cell-derived sEV and I would recommend its publication in the journal “cells”.
Author Response
Response to Reviewer #1
Q1. It should be noted and discussed in the paper that the sampling of total T cell population in the circulation is expected to approach a mixture of populations, probably mostly naïve cells due to their abundance. Are there any published data on sEV production that compare naïve, memory and activated T cells?
A1. To the best of our knowledge, the only published data concerning sEVs produced by subpopulations of T cells involve sEVs produced by CD4+ T cells (sEV mediate co-stimulatory functions), CD8+ T cells, activated CD8+ T cells (sEV mediate suppressive functions), and by the Treg subset (sEV are strongly immunosuppressive). There are no data available on sEv produced by naïve or memory T cells. However, neither study addressed comprehensively proteome component of analyzed vesicles. This explanation was added to the revised manuscript along with the following pertinent references [27-29]:
For T cells in general: Shao Y, Pan X, and Rong Fu. Role and function of t cell derived exosomes and their therapeutic value. Mediators of Inflammation (2021); https//doi.org/10.1155/2021/ 8481013.
For activated T cells: Azoulay-Alfaguter I, Mor A. Proteomic analysis of human T cell-derived exosomes reveals differential RAS/MAPK signaling. Eur. J. Immunol (2018)48 (11)1915-1917; https://doi.org/10.1002/eji.201847655
For Treg-derived sEV: Tung SL, Boardman DA, Sen M, et al. Regulatory T cell-derived extracellular vesicles modify dendritic cell functions. Sci. Rep. (2018) 8(1):6065. Doi 10.1038/s41598-018-24531-8.
Q2. The authors describe several T cell “function” that are differentially expressed between the two groups of sEV. It would be helpful to mention if any specific, typical proteins will be mentioned as well, for additional validation of the data.
A2. Thank you for this remark. To address this comment in the revised manuscript, the set of proteins identified in CD3(+) exosomes was compared to the set of proteins identified in CD3+ T-cells. The most comprehensive description of the T-cell proteome currently available in the literature is the work of Joshi RN et al (2019) (newly added reference [26]). The authors performed an in-depth proteomic analysis of T-cells described as CD3+/CD4+/CD8- T-cells (6572 proteins were identified). The new Venn diagram provided in the revised Figure 2A represents the overlap between proteins identified in T-cells (according to Joshi et al.) and 263 proteins characteristic for CD3(+) exosomes (i.e., proteins that were neither putative plasma proteins nor proteins overexpressed in CD3(-) exosomes). iT is noteworthy that the majority of proteins characteristic of CD3(+) exosomes were expressed in CD3+ T cells. On the other hand, the majority of 92 exosome proteins not detected in T cells were associated with exosome-based transport and putatively represent components specific for extracellular vesicles (this data is provided in a new Supplementary Figure S1).
Reviewer 2 Report
In this manuscript, Aneta Zebrowska et al. present data on proteomic and metabolomic profiling T cell-originated exosomes circulating in human blood. The manuscript is focused on the comparison of composition of small extracellular vesicles separated from healthy donor plasma using of CD3 monoclonal antibody with the composition of CD3-negative sEV fraction from the same source. Authors collected valuable information regarding the protein and lipid content in these two fractions. The CD3-based approach for T cell-derived exosome isolation was described in general in the authors’s earlier works.
This study drives attention to the potential use of human plasma samples as a ”liquid biopsy” of T cells, in other words, to analyze variations in the exosome membrane and cargo content. The main limitation of the work is that no data on the protein composition for individual donors are presented in the manuscript. It is the data that could give an idea of the similarity or differences of the composition in different donors. This issue is not discussed in the manuscript.
There is a question to the study, why more than 50% of the references cited in the manuscript are the papers published, at least in part, by the same authors. It may be caused partially by the originality of the study, still the topic concerning the isolation, identification, proteomic analysis of exosomes of leukocyte origin is obviously rather wide and popular and is worthy of a more extensive bibliography.
Minor concerns
Centrifuge speed is indicated in some cases in rpm. Using g is preferred.
It is better can be changed “minutes” for “min” throughout the text.
There are some inaccuracies in the description of companies.
Author Response
Response to Reviewer #2
Q1. The main limitation of the work is that no data on the protein composition for individual donors are presented in the manuscript. It is the data that could give an idea of the similarity or differences of the composition in different donors. This issue is not discussed in the manuscript.
A1. Assessment of interpersonal differences between individual donors was outside the scope of our study, therefore it was not discussed in the manuscript. Instead, we focused on differences in pairwise compared exosome fractions from the same donor. However, following this suggestion, we have added additional information to Supplementary Table S1 including normalized abundance of proteins identified in CD3(+) and in CD3(-) exosomes from plasma of individual healthy donors, which enables assessment of such differences if necessary.
Q2. There is a question to the study, why more than 50% of the references cited in the manuscript are the papers published, at least in part, by the same authors. It may be caused partially by the originality of the study, still the topic concerning the isolation, identification, proteomic analysis of exosomes of leukocyte origin is obviously rather wide and popular and is worthy of a more extensive bibliography.
A2. We agree that numerous papers exist concerning the isolation of exosomes and analysis of their proteome content in general. However, isolation of exosome populations derived specifically from T cells is a novel approach that is just becoming recognized. It is not yet popular and there are only a few relevant publications in the literature. We have added citations of other authors in the text (i.e. [27-29]. Also, see Answer A1 for Reviewer #1.
Q3. Minor concerns: Centrifuge speed is indicated in some cases in rpm. Using g is preferred. It is better can be changed “minutes” for “min” throughout the text. There are some inaccuracies in the description of companies.
A3. Thank you very much for pointing out inaccuracies in the text. The centrifuge speed set in “rpm” (section 2.5) was incorrect and it should be “rcf” (or “g”). Hence, we corrected the value to 20,000xg in the manuscript text to keep it consistent with other centrifugation speed values given in Materials and Methods. As for the remaining comments: “minutes” were changed to “min” throughout the text. Description of companies was checked and missing information was completed.
Reviewer 3 Report
The manuscript is clear, concise, and well written. However, the authors should be more cautious about their statement on the utility of CD3+ exosomes as "T cell biopsy". The data they provide are indeed suggestive, but not conclusive, as no direct comparisons have been done. To overcome this limitation, they should compare the relative abundance of the substance in CD3+ exosomes with matched T cell samples. Another limitation is related to the sample size: 10 cases are not suffiicient to make general statements.
The authors are encouraged to perform further experiments to support their points. Differently, these limitations should be acknowledged and discussed in the appropriate sections (eg the last paragraph of the Results and Conclusions).
Author Response
Response to Reviewer #3
Q1. (…) the authors should be more cautious about their statement on the utility of CD3+ exosomes as "T cell biopsy". The data they provide are indeed suggestive, but not conclusive, as no direct comparisons have been done. To overcome this limitation, they should compare the relative abundance of the substance in CD3+ exosomes with matched T cell samples. Another limitation is related to the sample size: 10 cases are not sufficient to make general statements. The authors are encouraged to perform further experiments to support their points. Differently, these limitations should be acknowledged and discussed in the appropriate sections (eg the last paragraph of the Results and Conclusions).
A1. Thank you very much for this valuable remark. In the revised manuscript the set of proteins identified in CD3(+) exosomes was compared to the set of proteins identified in CD3+ T-cells. The most comprehensive description of the T-cell proteome currently available in the literature is the work of Joshi RN et al (2019) (newly added reference [26]). The authors performed an in-depth proteomic analysis of T-cells described as CD3+/CD4+/CD8- T-cells (6572 proteins were identified). The new Venn diagram provided in the revised Figure 2A represents the overlap between proteins identified in T-cells (according to Joshi et al.) and 263 proteins characteristic for CD3(+) exosomes (i.e., proteins that were neither putative plasma proteins nor proteins overexpressed in CD3(-) exosomes). This is noteworthy that the majority of proteins characteristic of CD3(+) exosomes were expressed in CD3+ T cells. On the other hand, the majority of 92 exosome proteins not detected in T cells were associated with exosome-based transport and putatively represent components specific for extracellular vesicles (this data is provided in a new Supplementary Figure S1).
[26] Joshi RN, Stadler C, Lehmann R, Lehtiö J, Tegnér J, Schmidt A, Vesterlund M. TcellSubC: An Atlas of the Subcellular Proteome of Human T Cells. Front Immunol. 2019;10:2708. doi: 10.3389/fimmu.2019.02708.
In our study, blood samples of 10 donors were selected. Hence, we would like to emphasize that the analysis of interpersonal differences among individual donors of CD3(+) exosomes was out of the scope of the current work, which would require a larger set of samples. Instead, our proof-of-the-concept study was focused on methodological approaches and testing general differences between CD3(+) and CD3(-) vesicles. Nevertheless, the presence of T-cell-derived proteins was confirmed in the CD3(+) exo fraction from all donors.
Hence, new data provided in the revised manuscript further supports the idea of the potential utility of CD3(+) exosomes as a “T-cell biopsy”. As suggested, data that support our conclusions are now included in the revised Results and Discussion.
Reviewer 4 Report
The manuscript by Aneta Zebrowska et al. describes the proteomic and metabolomic profiles of T cell-derived exosomes isolated from human plasma. This manuscript should supply more evidence to support the conclusions. In my view, this manuscript needs minor revision before publication in MDPI, cells.
Author Response
Response to Reviewer #4
Q1. This manuscript should supply more evidence to support the conclusions. In my view, this manuscript needs minor revision before publication in MDPI, cells.
A1. In response to the Reviewers’ comments, we have modified the manuscript providing new evidence to support our conclusions. The most important changes include: (1) we have compared the set of proteins identified in CD3(+) exosomes with proteins detected in T-cells (revised Figure 2); (2) we provided abundance of proteins identified in CD3(+) and CD3(-) exosomes from plasma of individual donors to illustrate their inter-individual variability (revised Supplementary Table S1); (3) we have performed an additional GO term analysis of proteins detected in CD3(+) exosomes (new Supplementary Figure S1); (4) we have added new pertinent references on sEVs produced by T cells. We hope that the introduced changes will satisfy Reviewer #4 as he/she did not list any specific shortcomings.
Round 2
Reviewer 2 Report
I was unable to find the Supplementary Figure 1 among the submitted documents. It would be great to look at it to finalize the reviewing process. Otherwise, I am satisfied with the revised version of the manuscript (they may just remove the extra hyphen at the line 229, page 5).
Reviewer 3 Report
The authors expanded their manuscript by adding further results to support their conclusions. Also, the supplementary tables are appropriate.
Overall, this revised version has been substantially improved. I have no further suggestions.